# Molecular Doping of CVD-Graphene Surfaces by Perfluoroalkyl-Substituted Perylene Diimides Derivatives

**DOI:** 10.3390/nano12234239

**Published:** 2022-11-28

**Authors:** Federico Chianese, Lucrezia Aversa, Roberto Verucchi, Antonio Cassinese

**Affiliations:** 1Dipartimento di Fisica, Università Degli Studi di Napoli Federico II, Piazzale Tecchio 80, 80125 Napoli, Italy; 2CNR-SPIN, Unità di Napoli, Piazzale Tecchio 80, 80125 Napoli, Italy; 3Institute of Materials for Electronics and Magnetism, CNR-IMEM, FBK Trento Unit, Via alla Cascata 56/C, 38123 Trento, Italy

**Keywords:** graphene, molecular doping, perylene, XPS, UPS, SKPFM

## Abstract

Non-covalent π-π and dipolar interactions with small aromatic molecules have been widely demonstrated to be a valid option to tune graphene work functions without adding extrinsic scattering centers for charge carriers. In this work, we investigated the interaction between a CVD-graphene monolayer and a thermally evaporated sub-monolayer and the following few-layer thin films of similar perylene diimide derivatives: PDI8-CN2 and PDIF-CN2. The molecular influence on the graphene work function was estimated by XPS and UPS analysis and by investigating the surface potentials via scanning Kelvin probe force microscopy. The perfluorinated decoration and the steric interaction in the early stages of the film growth determined a positive work function shift as high as 0.7 eV in the case of PDIF-CN2, with respect to the value of 4.41 eV for the intrinsic graphene. Our results unambiguously highlight the absence of valence band shifts in the UPS analysis, indicating the prevalence of dipolar interactions between the graphene surface and the organic species enhanced by the presence of the fluorine-enriched moieties.

## 1. Introduction

As a material, graphene harbors remarkable qualities that justify the restless research activity of the last two decades. Its hexagonal carbon monolayer structure exhibits a superior electronic quality such that charge carriers can travel ballistically at the micrometer scale with hole and electron mobilities of more than 10^5^ cm^2^ V^−1^s^−1^ [1], in addition to nearly ideal mechanical and thermal properties [2,3]. Several applications have been reported so far in the fields of electrocatalysis [4], electrochemical sensors [5], and energy storage devices [6]. Graphene’s two-dimensional nature yields ultra-low wavelength-independent optical absorptions for normal incident light below 3 eV [7], creating strong interest in its use as novel electrode material in transistors, light-emitting diodes, solar cells, or as a plasmon-enhanced absorber material in optoelectronic devices [8,9,10,11]. With their intrinsic semi-metallic nature, graphene films are expected to promptly close the technological gap in more common transparent electrodes, such as ITO, AZO, and other carbon-based alternatives, in terms of sheet resistance, resiliency to mechanical stress, and optical properties [12].

This framework has given particular attention to its application as a source-drain electrode in organic electronic devices [13]. Several experimental works have exploited the advantages of graphene for controlling, for example, the morphological mismatch at electrode–organic interfaces [14,15] or in alleviating short channel effects in nanometric organic field-effect-transistors (OFETs) [16,17]. The use of graphene electrodes for enhancing contact injection in OFETs has been attributed, among other things, to the field-effect tunability of its work function [18], resulting in direct control over the injection barrier at the heterointerfaces [19,20].

Given the versatility of graphene when applied as an electrode material, precise control over its doping state is an absolute necessity for tailoring the transport properties and optimizing its integration into a specific device’s architecture. Namely, the doping conditions of graphene are given by the relative position of the Fermi energy (E_F_) above (*n*-type) or below (*p*-type) the Dirac point (E_D_ ≈ 4.5 eV). The magnitude of the E_F_ shift controls the static excess of holes or electrons without external field-effect gating. This directly impacts the intrinsic conductivity of a graphene sheet and its effective energetic compatibility with an active material in the final device. Graphene is a two-dimensional material sensitive to virtually any intentional and unintentional exposure to external contamination [21]. Its electronic properties have been efficiently tuned with various approaches such as atomic and molecular physisorption [22,23], heteroatom substitution [24,25], or covalent decoration [26].

Surface functionalization with polymeric and molecular layers can reliably control graphene’s *n*-type and *p*-type doping. In this case, aromatic molecules can stably bind to single-layer graphene films through strong π-π stacking or pure dipolar interactions that mainly preserve the homogeneity of a honeycomb lattice and its transport performances. The *n*-type doping is primarily obtained through interactions with aromatic molecules having electron-donating groups, such as 1,5-naphthalenediamine (Na-NH2) [27], vanadylphthalocyanine (VOPc) [28], or molecular reductant ruthenium-based dimer ([RuCp*Mes]_2_) [29]. On the contrary, organic moieties such as tetrasodium 1,3,6,8-pyrenetetrasulfonic acid (TPA) [27] and para-hexaphenyl (6P) [30] have been reported to induce p-doping by accommodating the extra electrons transferred from the graphene surface towards the electron-withdrawing moieties. In this regard, tetrafluorotetra-cyanoquinodimethane (F4-TCNQ) appears to be one of the compounds extensively used as a benchmark for the environmentally stable modification of graphene’s work function. According to combined DFT calculations and high-resolution photoemission spectroscopy (PES), F4-TCNQ induces a shift of approximately 1.4 eV of the graphene work function [31,32]. Angle-resolved PES experiments have partially elucidated the role of the electronically active cyano groups (C≡N) in the doping processes and highlighted the enhancement of the E_F_ shift as being electrostatically mediated by the presence of fluorine (F) atoms [33]. Despite the reported results, a thoughtful analysis comprising the role of the organic thin film morphology, molecular clustering, and local organic–graphene interaction has not fully been explored.

This work will follow this path, investigating in more detail the role of the fluorine-rich molecular end groups over the doping state of the CVD-graphene surfaces. For the scope, we chose two perylene tetracarboxy diimides derivatives (PDI8-CN2 and PDIF-CN2) that have widely been employed as top-performing environmentally stable *n*-type organic semiconductors [34]. The two materials mainly differ from the molecular substitution of the side chains, i.e., perfluoro alkyl substitution (CH_2_-C_3_F_7_), for the PDIF-CN2 against the alkyl chain-decorated PDIF8-CN2 (C_8_H_17_). Both materials share the same electron-withdrawing cyano insertions on the perylene cores. We probed the graphene work function shift and molecule–graphene interaction with PES analysis, supported by an investigation of the surface potential maps retrieved via scanning Kelvin probe force microscopy (SKPFM). This latter technique is based on determining the contact potential difference between a conductive AFM tip and a sample surface with high spatial resolution, unraveling the additional role of the molecular ordering over the total electrostatic interaction at a nanoscopic level.

## 2. Materials and Methods

Samples were fabricated starting from commercial monolayer CVD-graphene (Graphenea, San Sebastian, Spain) and transferred onto SiO_2_(300 nm)/Si multilayered substrates. PDI8-CN2 and PDIF-CN2 (FlexTerra N1100 and N1200, respectively, Skokie, IL, USA) were evaporated from powder precursors via organic molecular beam deposition (OMBD) in high vacuum conditions (P ≈ 10^−7^ mbar). Deposition rates of *R* = 0.3 nm/min for PDIF-CN2 and *R* = 0.4 nm/min for PDI8-CN2, respectively, were employed. Before deposition, the graphene substrates were carefully cleaned in an acetone bath and rinsed with ethanol (Merck, Darmstadt, Germany). During the OMBD process, the substrates were kept at 100 °C via an external PID temperature control system. The organic thin film thickness was monitored through a quartz crystal thickness monitor (Intellemetrics IL150, Paisley, UK).

Photoelectron spectroscopies (PES) were carried out in an ultra-high-vacuum (UHV) system using a VSW HA100 hemispherical electron energy analyzer with a PSP power supply and control (Macclesfield, UK) [35]. UV photoelectron spectroscopy (UPS) was performed using an He I photon with the energy at ~21.2 eV, with a final energy resolution of 0.1 eV (1 mm diameter probing spot). The work function (WF) was calculated using the position of the secondary emission cut off (SECO), while the ionization potential (IP) was calculated using the spectrum total length. We performed X-ray photoelectron spectroscopy (XPS) with a non-monochromatized Mg Kα source (photon at 1253.6 eV), with a final energy resolution of 0.86 eV. The analysis system was not equipped with a charge compensator, and we did not observe energy shifts or peak broadening due to charging during analysis in this study. The binding energy (BE) scale for the UPS was evaluated from the Fermi energy level of an Au sputtered surface. In the case of the XPS analysis, we calibrated the data from the Au4f 7/2 at 84.0 eV of the same Au substrate. The core level analysis was performed by Voigt line-shape deconvolution after the background subtraction using a Shirley function. The typical precision for each component’s energy position was ±0.05 eV. The uncertainty for the full width at half-maximum (FWHM) was less than ±2.5%, while it was approximately ±2% for the area evaluation.

We inferred the samples’ morphologies and surface potential mappings via non-contact atomic force microscopy (NC-AFM) and amplitude-modulation scanning Kelvin probe force microscopy (AM-SKPFM) using a XE-100 Park Systems (Mannheim, Germany) AFM. The SKPFM technique is based on acquiring the contact potential difference (*V*_cpd_) between a sample and a conductive AFM tip when the two are brought into close contact (≈nm) [36,37,38]. The work function difference between the two different materials was translated in an electrostatic force component that added up to the Van der Waal attractive contribution perturbing the oscillation condition of the tip scanning the surface in non-contact mode. A voltage feedback loop counteracted the external force by applying an equal and opposite DC term to the tip that locally nullified the electrostatic contribution and restored its optimal interaction condition. The externally applied feedback voltage signal individuated the surface potential (*V_surf_*) through:(1)$$V_{surf}=(\phi_{tip}-\phi_{surf})/e$$
where $\phi_{surf}$ and $\phi_{tip}$ are the work function of the surface under analysis and the tip, respectively. For the reliable estimation of an unknown $\phi_{surf}$, $\phi_{tip}$ was preliminarily acquired by calibrating the Au-coated tip on a freshly cleaved highly ordered pyrolytic graphite (HOPG) sample. Considering the latter as a stable measurement standard having $\phi_{HOPG}$ = 4.65 eV [39], the work function of the surface under analysis could be easily retrieved via Equation (1). For the proposed experiments, the measured value for the tip was $\phi_{tip}=4.81\pm \;0.01\;\mathrm{eV}$.

The analysis was carried out in air using an XE-100 Park atomic force microscope respectively equipped with a PPP-NCHR by NanoSensors (Neuchatel, Switzerland) for standard NC topographies and NSC14 Cr/Au conductive cantilevers by MikroMash (Tallinn, Estonia) for the potential mapping via SKPFM. The potential profile signal was demodulated by employing an external Stanford Research System SR830 DSP Lock-in amplifier (Sunnyvale, CA, USA) using a sinusoidal reference with a frequency of 18 kHz and a *V*_AC_ amplitude of between 1.8 V and 1.9 V.

## 3. Results

### 3.1. Morphological Analysis

As a preliminary morphological analysis, we investigated the molecular nucleation of the submonolayer films (*t* = 1 nm) on the graphene substrate. For both materials (Figure 1a,b), the organic thin film growth germinated predominantly in alignment with the substrate defectivities, such as the graphene wrinkles, whose morphological features could be detected, as shown in Figure 1c. This effect was particularly evident in the case of the PDIF-CN2, for which sparser nucleation sites were typically observed if we considered equal nominal thicknesses for PDI8-CN2. The typical line profiles of PDIF-CN2 grains showed an average height of 3.3–3.6 nm (Figure 1e). Considering the organic molecule’s reported crystalline structure [40], such a value was incompatible with a triclinic unit cell, which was characterized by a nominal 1.95 nm d-spacing related to the long molecular axis. It was thus possible to invoke the presence of an initial wetting layer given by nearly flat molecules with a thickness of approximately 1.3 nm, which, in turn, suggested an enhanced interaction of the molecules with the graphene substrate, possibly those of the CN and CF groups, which came in closer contact with the carbon lattice.

The PDI8-CN2 case, on the other hand, showed morphological features which were more compatible with those observed in standard substrates, such as untreated SiO_2_ surfaces [41]. We observed molecular terraces of ≈1.8 nm for the sub-monolayer samples. Compared to the reported values retrieved from the X-ray diffraction spectra (d-spacing = 1.98 nm), this feature highlighted an enhanced tilt angle to the graphene surface. The morphological evolution for the increasing nominal thicknesses in PDIF-CN2 followed the first stages of nucleation, with the organic films rapidly covering the graphene surface according to a layer-plus-islands or Stranski–Krastanov growth mechanism (Appendix A). For PDI8-CN2, the formation of the islands seemed to be the driving mechanism in the very first growth stages, with an inter-molecular interaction being more critical than the organic-graphene one, as suggested by the observed aggregation scheme [41].

### 3.2. Photoelectron Spectroscopies

We obtained reference spectra of the organic molecules by analyzing thick films (25 nm) deposited on 200 nm SIO_2_/Si(100). The atomic percentages from the XPS analysis for PDI8-CN2 were C 83.2%, N 8.2%, and O 8.6%, which were in agreement with the theoretical values of 84.0%, 8.0%, and 8.0%, respectively. The C1 core level (Figure 2a, top curve) showed contributions from C−H/C=C bonds in the molecular backbone at 284.3/284.8 eV, from C-C lateral chains at 285.7 eV, from the cyano groups (−C≡N) at 286.9 eV, and from the imide groups (−N−C=O) at 288.5 eV, as well as a broad satellite peak at 290.2 eV due to the shake-up processes typical of π-conjugated [42,43]. Another peak was present at 286.2 eV and represented C atoms bonded to electron attractor groups (labeled as “other C”), i.e., close to the cyano and imide groups [44]. Analysis of the N1 core level (Figure 2b, top curve) revealed the presence of −C≡N and −N−C=O functional groups at 399.6 eV and 400.8 eV [42,44]. The two peaks’ intensity ratios were close to 1, as expected. The O1s core level (Appendix A, top curve) was characterized by a single component at 531.8 eV due to the imide groups.

PDIF-CN2 showed electronic properties different from those of PDI8-CN2. The atomic percentages were C 60.6%, N 6.4%, O 6.5%, and F 26.5%, in agreement with the theoretical values of 60.6%, 7.2%, 7.2%, and 25.0%, respectively. The C1s core level (Figure 3a, top curve) showed the contribution from the aromatic backbone shifted at 285.8 eV, a higher BE compared to PDI8-CN2 (as observed in case of F substitution in lateral chains [45,46] and of other electron attracting groups [47,48]), and a larger FWHM (1.81 instead of 1.27 eV) due to merging of the C=C and C−H contributions. The other peaks were related to the cyano and imide groups (286.8 eV and 288.7 eV, respectively) and the −CF2 and −CF3 groups (291.1 eV and 293.6 eV, respectively) [49]. The N1s (Appendix A, top curve) and O1s (Appendix A, top curve) core levels showed the same features identified for PDI8-CN2 (within ±0.1 eV, see Appendix A). As expected, two components related to the fluorinated functional groups, CF2 at 689.1 eV and CF3 at 688.4 eV, were present in the F1s photoemission peak (Figure 3b, top curve), with a 4/3 intensity ratio.

We performed PES analysis on the PDI8-CN2 and PDIF-CN2 thin films deposited on graphene at nominal thicknesses of 2, 4, and 8 nm, and 2, 3, and 10 nm, respectively. The long-range spectra (Appendix A) showed the presence of all expected elements, i.e., C, N, and O for both organics and F for PDIF-CN2. Stoichiometry in all films resembled the complete molecular film data, suggesting no organic degradation occurred during the deposition process. The C1s and O1s core levels for the bare graphene substrate are shown in Appendix A. We identified three components related to the C=C−C main sp2 graphene structure (284.77 eV), the C−C sp3 bonds (285.55 eV, defects), and the C−O bonds (286.92 eV). The O1s main component at 533.04 eV was due to the SiO_2_ substrate, while the peak at 533.22 eV was related to the C−O groups [50]. Although we found the presence of a C1s C−O peak, this was likely related to the graphene defects created in the transfer process from the CVD synthesis, and our substrate had completely different characteristics from the graphene oxide [50].

#### 3.2.1. PDI8-CN2 Film Growth

In the case of the PDI8-CN2 thin films, the C1s core level (Figure 2a) appeared as a complex and broad peak. It could be reproduced as the superposition of contributions from the graphene substrate and the growing organic layer, according to our previous results for the thick films. The intensity of the PDI8-CN2 C1s components increased with film thickness, without the appearance of new features. On the contrary, the intensity of the graphene C1s signal for 2 nm film was 59.8% of the total C1s core level area, and it slowly decreased with the increasing thickness and remained 21.1% at 8 nm (see Table 1). This suggested a non-uniform coverage, with the formation of islands in the early growing stage slowly increasing their area, in agreement with AFM results. In agreement with AFM results, molecule–molecule pairing, rather than a molecule–surface interaction, was the primary process. We did not observe significant BE shifts (within ±0.1 eV) and FWHM changes (see Appendix A) at the different film thicknesses. The O1s spectra (Appendix A) showed the main contribution from the SiO_2_ substrate, two additional components related to the imide group in PDI8-CN2, and CO from the graphene substrate. The N1s core levels showed the two known peak structures (Figure 2b), having almost constant BEs for the 4 and 8 nm films. For the 2 nm film, the cyano contribution had a lower weight on the total area, with a 0.9 intensity ratio with the imide peak. A peak broadening in the 397–399 eV region appeared and could be reproduced by introducing a new component at 398.28 eV, labeled CN*. The role of the −C≡N groups in mediating a weak interaction with the graphene had already been proposed in the case of F6TCNNQ [51], with an electron accumulation on the cyano groups at very low coverage and BE shifting at a lower BE of the N1s-related component. In our case, the −C≡N peak intensity was reduced, and the theoretical ratio with the imide component was restored when we also considered the weight of the new CN* peak. A previous study of PDI8-CN2 on Au electrical conductance showed similar results: the cyano functional groups drove the chemical interaction at the interface processes [52]. We concluded that the absence of significant changes in BE for all the core levels suggested a weak molecular interaction with the graphene substrate, with the role of the −C≡N groups being mediation of this interface process.

#### 3.2.2. PDIF-CN2 Film Growth

The results for the PDIF-CN2 film growth revealed a more complex scenario. Figure 3a shows the C1s core level for the three films, and it was analyzed as the superposition of contributions from the substrate and the growing organic layer. We observed a rapid decrease in the graphene C1s intensity with the film thickness, with a residual contribution to the total C1s signal of 6.2% at 10 nm (see Table 1). This suggested a different growing mechanism from for PDI8-CN2, with higher molecules sticking to the surface and a homogeneous and complete final coverage, which was in agreement with AFM results.

The C1s molecular components shifted to lower BEs at the highest coverage of 10 nm, approximately −0.15 eV to the 2 and 3 nm films, while the graphene components remained almost unchanged. N1s and O1s did not show significant shifts, possibly due to the low signal intensity. However, the F1s core level showed a similar −0.2 eV BE shift (see Appendix A), suggesting an interaction at the interface involving the whole molecule rather than some specific groups. The N1s core level showed the presence of a third component, approximately 1.2 eV away from the cyano peak, as already observed for the PDI8-CN2 film at 2 nm (see Figure 2b), and it was consequently labeled as CN*. It was likely evidence of the interaction of some cyano groups with the graphene layer [51,52]. The BE shifts suggested a different and more complex scenario from PDI8-CN2 and will be discussed later.

### 3.3. UPS

The valence band photoemission spectra for PDI8-CN_2_ are shown in Figure 4a, and they outline the evolution from low to high coverages. The spectrum for the thick film is also shown (labeled as REF in Figure 4a). Molecular bands were superimposed on graphene bands with a fixed binding energy position (HOMO at 2.85 eV) for all film thicknesses [53]. The graphene features were still detectable at the higher coverage, confirming the presence of PDI8-CN2 islands and incomplete substrate coverage. The WF was almost stable at 4.77–4.79 eV for all films (see Figure 4a and Table 2), approximately +0.35 eV from the experimental graphene value (at 4.41 eV). The IP showed a similar trend, with a 7.0 eV value for all films. The absence of significant BE changes in the MO positions (and in the main core levels) suggested the presence of a +0.6 eV dipole at the interface. The weak interaction of the cyano groups with the graphene was unlikely to produce detectable band-bending phenomena. Moreover, the WF and IP values were very similar to those of the PDI8-CN2 thick film at approximately 4.92 and 7.10 eV [53,54] (see Table 2), confirming the formation of well-structured perylene islands right from the early growth stages. Considering a HOMO-LUMO gap of approximately 2.5 eV [53,54], the PDI8-CN2 on the graphene system showed marked *n*-type semiconducting properties.

PDIF-CN2 valence bands (Figure 4b) showed the typical organic molecular orbitals (MOs), clearly distinguishable right from the 2 nm film, while the graphene bands completely disappeared at the highest coverage. This agrees with the homogeneous and complete substrate coverage typical of Stranski–Krastanov growth, as previously described. The HOMO was located at 2.9–2.7 eV and HOMO-1 was at 4.3–4.5 eV, with a decreasing energy position at an increasing coverage (−0.12 eV at 10 nm vs. 2 nm). The WF was constant at the 2/3 nm films and increased at 10 nm, from 4.93/4.96 eV (+0.52/0.55 eV vs. graphene) to 5.12 eV (+0.71 eV) (see Figure 4b and Table 2). The IP showed a similar trend, from 7.09/7.12 to 7.38 eV. Considering the WF and IP values of 5.38 and 7.33 eV, respectively, for the thick PDIF-CN2 films (labeled as REF in Figure 4b) [55], a well-structured organic film was achieved only at 10 nm, confirming the Stranski–Krastanov growth. Similar to PDI8-CN2 [53,54], the PDIF-CN2 on the graphene system showed *n*-type semiconducting properties. The bands (as well the core levels) energy shifts suggested the presence of band bending, possibly with charge accumulation at the interface, but no new structures due to LUMO orbital filling were present in the HOMO-LUMO gap. The −0.12 eV shift was achieved only at the highest film thickness (see Table 2), and so it was likely related to the formation of PDIF-CN2 islands with the molecules having a different orientation to the interface layer, giving rise to a weak but detectable band-bending. In the early growth stages, the molecules lay on the graphene surface, allowing the interaction of both perfluoroalkyl chains with the substrate and creating a dipole of approximately −0.55 eV. The second layer showed standing molecules, with a weak charge transfer toward the first wetting layer inducing a further 0.16 eV WF variation (see Table 2), thus suggesting a complex intermolecular interaction before the completion of the organic film.

### 3.4. SKPFM Analysis

The surface potentials acquired via SKPFM in air are reported in Figure 5. The potential maps appeared to be substantially different between the two materials from the early growth stages (2 nm). For PDIF-CN2, the darker spots identified the islands on the brighter exposed graphene surface (Figure 5d). The surface potential evolution as a function of the nominal thickness for PDIF-CN2 can be seen in Figure 5e,f. For the 3 nm film, the potential maps showed the presence of multiple rectangular spots surrounded by exposed graphene, indicating the local de-wetting of the surface and the subsequent formation of organic crystallites (typically 1 µm long).

The individual features of the organic islands and those of the graphene substrate are lost in the 10 nm film (Figure 5f), where a complete molecular film was achieved. Conversely, for the PDI8-CN2 thin films, the single nucleation sites at 2 nm were barely discernible from the surface potential map (Figure 5a), with low potential contrast between the organic material and the graphene substrate. Extended darker areas could be tentatively explained in terms of substrate inhomogeneities, i.e., electron-hole “puddles” deriving from the interaction of graphene with SiO_2_ or from intrinsic morphological disorder [56]. Lastly, the thicker PDI8-CN2 films (Figure 5b,c) showed granular features that could be ascribed to the organic material’s multilayered structure, i.e., grain boundaries, rather than to actual interaction with the substrate.

Considering the single perylene/graphene potential maps of Fiugre 5, the statistical distributions of the single pixels from the surface potential maps (see Figure 6a,b) allowed us to estimate the vacuum level shift at the heterointerface via Equation (1). In the case of the PDIF-CN2/graphene heterointerface, the surface potential contributions were well separated as a direct consequence of the enhanced potential contrast. In this sense, it was possible to directly assess the graphene surface’s doping level, which was not covered by the organic overlayer. A typical difference of Δ≈100 meV was retrieved considering a WF of 5.19 eV for the graphene and 5.29 eV for the PDIF-CN2 layer for t = 2 nm (Figure 6a). By considering their evolution with thickness, PDIF-CN2 still showed two distinct peaks at 3 nm with a difference of ≈150 meV. The contribution due to the organic material appeared to saturate towards 5.33 eV for the thicker films (10 nm) when the double-peaked feature was lost due to the complete PDIF-CN2 film coverage of the graphene surface. Conversely, in the case of the submonolayer PDI8-CN2 films (Figure 6b), the two contributions from the graphene substrate and the organic thin film appeared to be nearly convoluted. A difference of approximately Δ ≈ 40 meV was calculated considering the graphene substrate WF (4.99 eV) and that of the organic surface (5.03 eV). For PDI8-CN2, the graphene contribution was rapidly lost from 3 nm, while the contribution due to the organic layer shifted progressively towards −5.17 eV at 8 nm. The overall WF evolution with the thicknesses is summarized in Figure 6c for both materials.

## 4. Discussion

Numerous remarks stem from crossing the XPS/UPS spectra and the potential landscapes obtained via SKPFM analysis. Firstly, the enhanced interaction in the case of PDIF-CN2 highlights the role of the perfluoroalkyl substitutions and, more specifically, the C-F bonds characterizing the PDIF-CN2 terminations as highly polarized moieties, given the extreme electronegativity of fluorine atoms [57]. This peculiar feature can tentatively also explain the flat steric interaction observed in the early stages of the thin film growth (sub-monolayer of Figure 1a) and, thus, the increased exposure of the graphene surface to the fluorine atoms. In a complementary fashion, holes are intrinsically accumulated on the graphene side, as schematically depicted in Figure 7a, resulting in the consequent shift of the work function with the formation of a surface dipole, as suggested by the UPS analysis.

Specifically, the graphene WF moves downwards with respect to the Dirac point up to 5.19 eV when interacting with the PDIF-CN2, i.e., a 0.39 eV difference if compared with the SKPFM analysis performed in air on a bare graphene surface (black dashed line in Figure 7b) and a 0.52 eV difference to the values deduced from the UPS spectra acquired in the UHV conditions (dotted line in Figure 7b). This also highlights the role played by the atmospheric contaminants (mostly H_2_O, O_2_, and CO_x_ molecules) in the determination of the energy level alignment at the heterointerface. The low difference between the SKPFM and the UPS values suggests the perfluoroalkyl chains prevent, to a great extent, the intrusion or the adsorption of contaminants [58], resulting in a steeper saturation of WF vs. the organic film thickness. The UPS and XPS analysis suggest the presence of a weak band-bending (approximately 0.12 eV) for the 10 nm vs. 3 nm film, in agreement with a 0.1 eV WF increase for the organic layers evidenced by the SKPFM analysis and likely related to the de-wetting process.

In the case of PDI8-CN2, the XPS/UPS analysis pointed out a milder interaction carried out predominantly by the electronegative C≡N groups, with the formation of a surface dipole that was nearly constant at all film thicknesses. Surface potential maps further confirmed this weaker interaction for PDI8-CN2 in terms of a potential difference of only ≈40 meV in the submonolayer case and a reduced doping state of the graphene substrate (4.99 eV blue dashed line in Figure 7b). The WF organic layer increased <0.1 eV from 2 to 3 nm and from 4 to 8 nm; thus, it could hardly be evidenced by UPS analysis.

Our graphene doping approach with perylenes led to reliable and promising results in terms of electronic level control compared with other organic molecules, as shown in Table 3.

## 5. Conclusions

We analyzed the role of two perylenes tetracarboxy diimides derivatives (PDI8-CN2 and PDIF-CN2) in tuning monolayer graphene electronic properties. Using photoelectron spectroscopies and scanning Kelvin probe force microscopy, we investigated the film growth mechanisms, the interface chemical features, and the work function vs. organic film thickness. PDI8-CN2 showed a typical island growth, with molecules tilted on the graphene surface with a mild organic–inorganic interaction, likely modulated by the organic cyano groups. We evidenced the formation of a surface dipole for all film thicknesses, leading to a WF increase of approximately +0.4 eV (at 8 nm) from the pristine graphene and *n*-type semiconducting properties. The PDIF-CN2 film showed a Stranski–Krastanov growth type, with a strong organic–graphene interaction due to the cyano and fluorinated groups, thus involving the whole molecule. The initial formation of a wetting layer (with molecules laying on the graphene) led to subsequent organic island growth, also with evidence of a de-wetting mechanism at higher coverages. We found the presence of a surface dipole (approximately +0.5 eV), while a weak band-bending (0.1–0.2 eV) occurred at the highest coverage of 10 nm and was likely related to the differently tilted orientations of the molecules.

Our results highlight the role of the perfluoro-alkyl substitution in unambiguously favoring a dipolar graphene–molecule interaction, which significantly tuned the inorganic electronic properties towards *p*-type doping conditions. In-vacuum and air analysis techniques led to very similar qualitative results, even if we found a nearly constant quantitative gap between the work function values, likely related to the different working conditions. The stability, diffusion, and possibility to tune the perylenes’ chemical groups make these organic molecules promising for controlling graphene electronic properties, given the possible industrial scale-up and exploiting the OMBD deposition technique compatibility with in-vacuum large-scale production systems. To this aim, we intend to check our systems’ morphologic and electronic properties with time to verify their stability.

## Figures and Tables

**Figure 1 nanomaterials-12-04239-f001:**
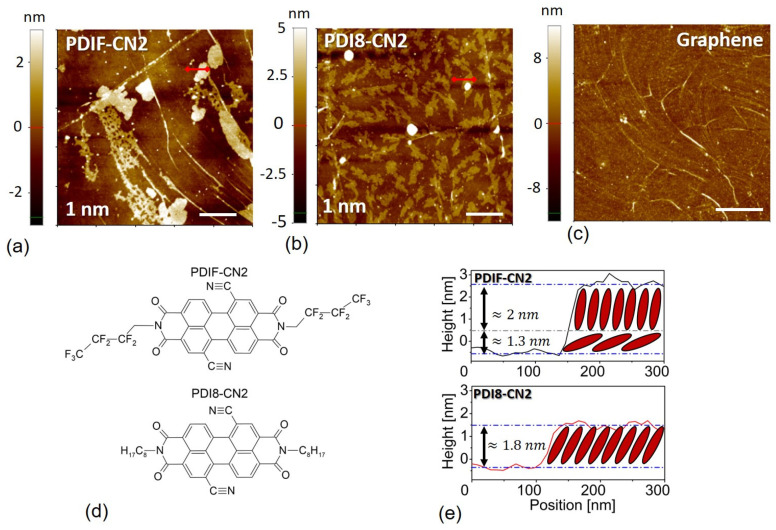
(**a**,**b**) The 5 nm × 5 nm NC-AFM topographies of the PDIF-CN2 and PDI8-CN2 sub-monolayers with nominal thicknesses of 1 nm, grown on the graphene surfaces. The white bars in the panels set a 1 nm reference. (**c**) Bare CVD-graphene surface as reference (10 µm × 10 µm, white bar a 2.5 µm reference), highlighting the sparse presence of wrinkles. (**d**) Molecular structures of the two perylene diimides derivatives employed in this work. (**e**) Line profiles corresponding to the red lines in panels (**a**,**b**) for the two materials. It is easy to deduce a nearly face-down molecular wetting layer (≈1.3 nm) in the PDIF-CN2 case.

**Figure 2 nanomaterials-12-04239-f002:**
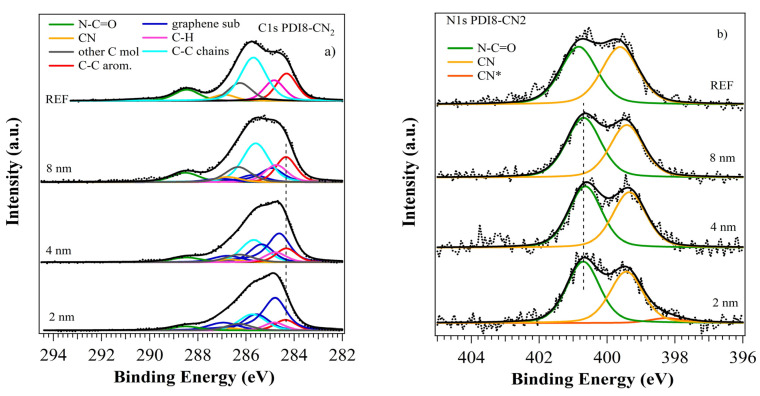
XPS analysis of the C1s (**a**) and N1s (**b**) core levels of the PDI8-CN2 film deposited on graphene, at different thicknesses. Spectra for the thick films are shown on the top for reference. The vertical lines suggest the absence of BE shifts for the main C1s and N1s molecular components.

**Figure 3 nanomaterials-12-04239-f003:**
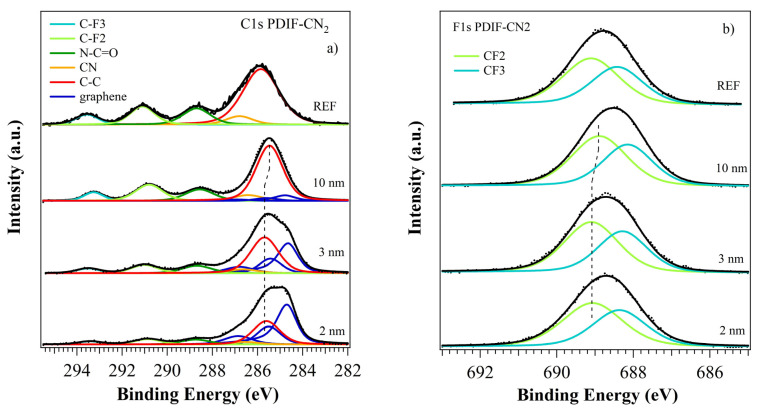
XPS analysis of the C1s (**a**) and F1s (**b**) core levels of the PDIF-CN2 film deposited on graphene, at different thicknesses. The spectra for the thick films are shown on the top for reference. The vertical lines suggest the BE shift for the main C1s and F1s molecular components for the 10 nm film.

**Figure 4 nanomaterials-12-04239-f004:**
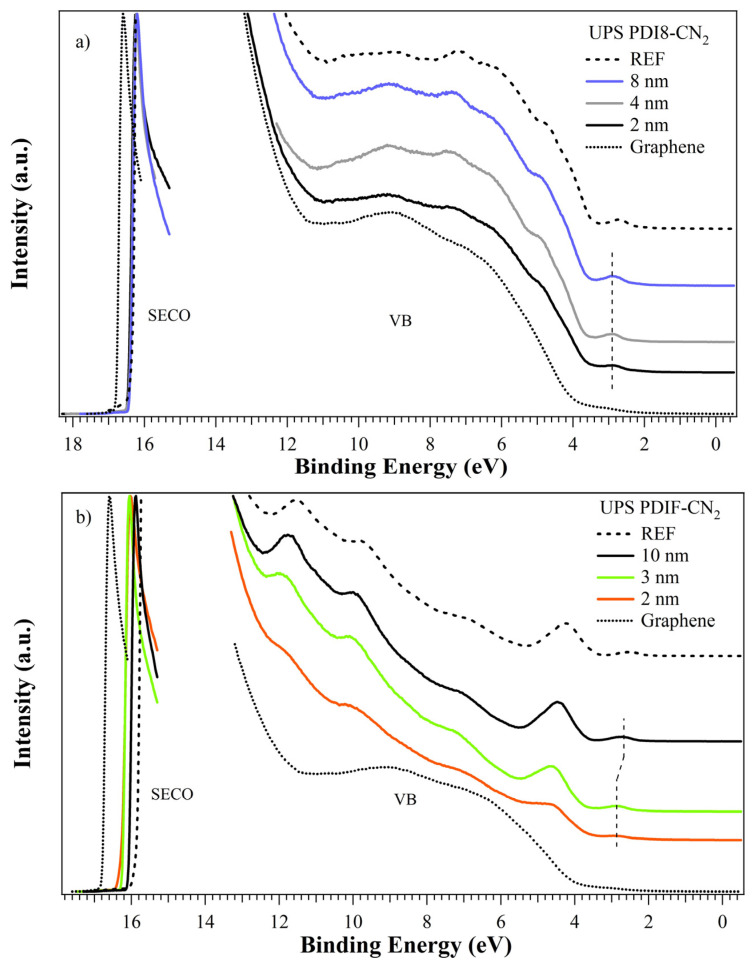
UPS spectra of the (**a**) PDI8-CN2 and (**b**) PDIF-CN2 thin films with different nominal thicknesses on CVD-graphene. Spectra for the thick film and the bare graphene are shown for reference.

**Figure 5 nanomaterials-12-04239-f005:**
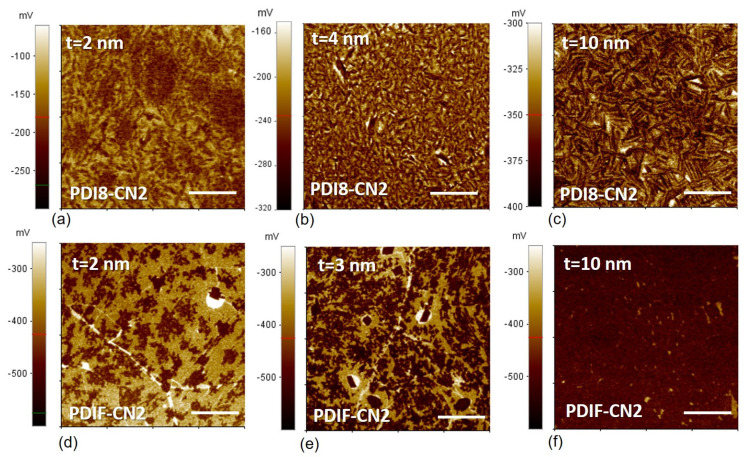
Surface potential maps (10 µm × 10 µm, 256 × 256 pxl) for the PDI8-CN2 (**a**–**c**) and for the PDIF-CN2 (**d**–**f**) grown on the graphene substrate, as a function of the nominal thickness t. The white bars correspond to 2.5 µm.

**Figure 6 nanomaterials-12-04239-f006:**
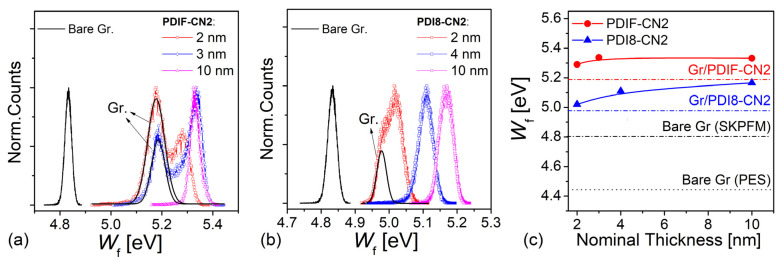
Statistical distributions of WF values retrieved from the calibrated SKPFM analysis for (**a**) PDIF-CN2 and (**b**) PDI8-CN2 considering the various organic thin film nominal thicknesses. The data are compared to the case of a bare graphene sample used as reference (black line). Panel (**c**) summarizes the evolution of the WF peaks for the organic semiconductors as a function of the thickness. The dashed-dotted lines individuate the graphene substrate contribution as retrieved from the sub-monolayer case (2 nm). The data are also compared to the WF of the bare graphene surfaces centered in the vicinity of 4.41 eV, as deduced by the UPS analysis in UHV conditions (black dotted line).

**Figure 7 nanomaterials-12-04239-f007:**
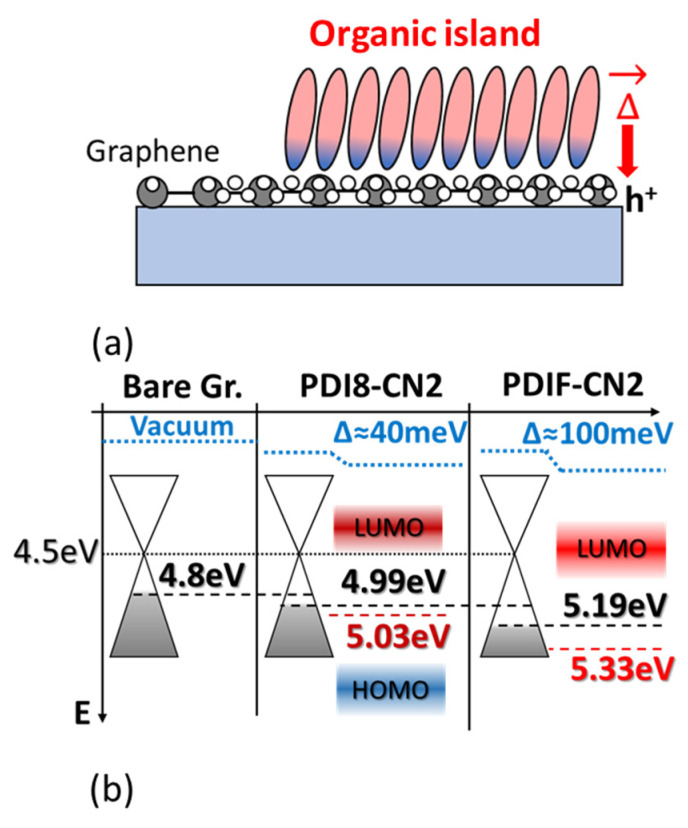
(**a**) Schematic depiction representing the charge accumulation phenomena occurring at the graphene–organic interface, resulting in an overall vacuum level shift of Δ. (**b**) Band diagram describing the energy level alignment at the graphene–organic interface, for both molecules, in the early stages of the organic thin film growth.

**Table 1 nanomaterials-12-04239-t001:** Intensity of the C1s signal from the graphene substrate on the total C1s photoemission peak for PDI8-CN2 and PDIF-CN2 at different film thicknesses.

	PDI8-CN2			PDIF-CN2		
	2 nm	4 nm	8 nm	2 nm	3 nm	10 nm
Graphene on C1s total signal (%)	59.8	49.9	21.1	54.3	38.2	6.2

**Table 2 nanomaterials-12-04239-t002:** Summary of the IP, HOMO, and WF values (from the UPS analysis) referred to in the PDI8-CN2 and PDIF-CN2 thin films with different nominal thicknesses. Values for the thick films and bare graphene are shown for reference.

Sample	IP (eV)	WF (eV)	HOMO (eV)
Graphene	-	4.41	-
PDI8-CN2 2 nm	7.04	4.79	2.85
PDI8-CN2 4 nm	7.02	4.77	2.87
PDI8-CN2 8 nm	7.04	4.79	2.85
PDI8-CN2 thick	7.10	4.92	2.71
PDIF-CN2 2 nm	7.09	4.93	2.87
PDIF-CN2 3 nm	7.12	4.96	2.87
PDIF-CN2 10 nm	7.38	5.12	2.75
PDIF-CN2 thick	7.33	5.38	2.54

**Table 3 nanomaterials-12-04239-t003:** Doping type and typical work function (WF) shift for the different organic molecules reported in the literature and in comparison with the results obtained in this work.

	Doping Type	WF Shift [eV]	Ref.
PDIF-CN2	*p*-type	≈0.5 (PES)/0.7 (SKPFM)	This work
PDI8-CN2		≈0.4 (PES)/0.5 (SKPFM)	
F4-TCNQ	*p*-type	≈1, 1.4	[31,33]
TCNQ		≈0.2	[33]
6P	*p*-type	-	[30]
[RuCp*Mes]_2_	*n*-type	1.8	[29]
VOPc	*n*-type	-	[28]
Na-NH_2_	*n*-type	-	[27]

## Data Availability

All data in this work are available on request by contact with the corresponding author.

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
