# Peer review of "Molecular Doping of CVD-Graphene Surfaces by Perfluoroalkyl-Substituted Perylene Diimides Derivatives"

_nanomaterials, 2022, doi:10.3390/nano12234239_

Round 1
Reviewer 1 Report
In this paper, the authors studied the interaction between CVD graphene monolayer and thermal evaporation sub monolayer and thin film of two perylene diimide derivatives (PDI8-CN2 and PDIF-CN2). The molecular influence on the graphene work function has been estimated by XPS and UPS analysis and by investigating surface potentials via Scanning Kelvin Probe Force Microscopy. I believe that publication of the manuscript may be considered only after the following issues have been resolved.
1. In order to better highlight the advantages of this work, the author needs to provide a table to compare related work.
2. The summary needs to be rewritten. The application background is mainly in the introduction, which cannot be expanded too much in the abstract. The summary part needs to focus on the work of this paper, and needs to mention the relevant significance of this work.
3. There should be a space between the number and unit of the full text, including the figure.
4. The introduction can be improved. The articles related to the some applications of graphene materials should be added such as Sensors 2022, 22, 6483; ACS Sustain. Chem. Eng. 2015, 3, 1677–1685; RSC Adv. 2022, 12, 7821–7829; Talanta 2015, 134, 435–442.
5. Please check the grammar and spelling mistakes of the whole manuscript.
Author Response
We thank the reviewer for all the interesting comments. Concerning all scientific questions, we answered each of them to improve paper quality and clarity. All modifications in the main text are evidenced in yellow. More in detail:
- In order to better highlight the advantages of this work, the author needs to provide a table to compare related work.
We thank the Referee for this valuable suggestion, indeed improving the possibility to compare our results with those presented in literature. As a reference list, we used the publications already inserted in our work. We put the new Table 3 in the Discussion section, adding the text: “Our graphene doping approach with perylenes leads to reliable and promising results in terms of electronic level control, if compared with other organic molecules, as shown in Table 3.”
- The summary needs to be rewritten. The application background is mainly in the introduction, which cannot be expanded too much in the abstract. The summary part needs to focus on the work of this paper, and needs to mention the relevant significance of this work.
We thank the Referee for this comment. Abstract has been modified reducing the state-of-the-art discussion and improving, emphasizing the results of our work. The new summary is:
“Non-covalent π-π and dipolar interactions with small aromatic molecules have been widely demonstrated to be a valid option to tune graphene work function without adding extrinsic scattering centers for charge carriers. In this work, we investigated the interaction between CVD-graphene monolayer and thermally evaporated sub-monolayer and few-layer thin films of similar perylene diimide derivatives: PDI8-CN2 and PDIF-CN2. The molecular influence on the graphene work function has been estimated by XPS and UPS analysis and by investigating surface potentials via Scanning Kelvin Probe Force Microscopy. The perfluorinated decoration and the steric interaction in the early stages of the film growth determined a positive work function shift as high as 0.7 eV in the case of PDIF-CN2, with respect to the value of 4.41 eV of intrinsic graphene. Our results highlight the absence of valence band shifts from the UPS analysis unambiguously, indicating the prevalence of dipolar interactions between the graphene surface and the organic species enhanced by the presence of the fluorine-enriched moieties.”
- There should be a space between the number and unit of the full text, including the figure.
We modified the main text and figures according to Referee’s suggestion.
- The introduction can be improved. The articles related to the some applications of graphene materials should be added such as Sensors 2022, 22, 6483; ACS Sustain. Chem. Eng. 2015, 3, 1677–1685; RSC Adv. 2022, 12, 7821–7829; Talanta 2015, 134, 435–442.
We thank the Referee for suggesting us these relevant papers. They have been inserted in Refs. 10, 4, 11, and 5 in the Introduction section, and properly commented.
- Please check the grammar and spelling mistakes of the whole manuscript.
We thank the Referee for the comment. We carefully checked typos and English language, also using programs like Grammarly (premium version).

Reviewer 2 Report
This interesting works deals with the moleculargrafting onton graphene layers to ajust its electron transpirt properties. The authors extensively studied their materials wiith a great data sown in main paper and supporting information. However this work should be carrefully revised before futher steps:
1) XPS results. Massive data is shown but the way authors did data curve fit is not appropriate:
Why C-C carbon has BE=285.7? It should be 285. The nitrile CN must be shifted from C-C by 1.8eV. Nitrogen from nitriles should be 399.6 eV. I suggest to check all positions and REFerences.
Why your graphene is so oxydized?
Please put C1s and O1s spectra for layers with thickness evolutions.
N1s position for imides seemed to be too high.
The shift of all peaks F1s and C1s can suggest that the charge compensation is not good.
2) Many references in the end appeards as errors. Fix it please.
3) Are planning to check the stability of your layers?
4) Any ideas about scaleup options of your methods? Please add a paragraph for industrual perspectives.
Author Response
We thank the Reviewer for all the interesting comments. We carefully checked typos and the English language, also using programs like Grammarly (premium version). All modifications in the main text are evidenced in yellow. Concerning all scientific questions, we answered each of them intending to improve paper quality and clarity. More in detail:
1) XPS results. Massive data is shown but the way authors did data curve fit is not appropriate:
Why C-C carbon has BE=285.7? It should be 285. The nitrile CN must be shifted from C-C by 1.8eV. Nitrogen from nitriles should be 399.6 eV. I suggest to check all positions and REFerences.
The Reviewer probably refers to the BEs for the PDIF-CN2 molecules. Differently from the PDI8-CN2 analysis, where different carbon-related species are distinguishable, we cannot identify the different C-C species for the fluorinated molecule, as suggested by the broad peak at 285.7 eV (FWHM 1.81 eV instead of 1.27 eV for PDI8-CN2, as claimed in line 218). In the literature are reported values for the main C1s peak ranging from 285 to about 286 eV. For molecules with electron attracting groups, such as CO, CN, imides, and CF, C1s main peak BE is shifted to values closer to 286 eV, as reported in these papers:
- Ref 44: Xray photoelectron spectroscopy analysis of hexafluorodianhydride–oxydianiline polyimide: Substantiation for substituent effects on aromatic carbon 1s binding energies; L. P. Buchwalter, B. D. Silverman, L. Witt, and A. R. Rossi, J. Vac. Sci. Technol. A 5, 226 (1987); doi: 10.1116/1.574108
- Structural and Electronic Properties of PTCDA Thin Films on Epitaxial Graphene; Han Huang, Shi Chen, Xingyu Gao, Wei Chen, and Andrew Thye Shen Wee; AcsNano, 3 (11), 3431, 2009
- Electronic properties of tetrakis(pentafluorophenyl)porphyrin Marco Nardi, Roberto Verucchi, Lucrezia Aversa,Maurizio Casarin, Andrea Vittadini, Nicola Mahne, Angelo Giglia, Stefano Nannarone and Salvatore Iannotta; NewJ.Chem., 2013, 37, 1036
- Line shapes and satellites in high-resolution x-ray photoelectron spectra of large p-conjugated organic molecules A. Scholl and Y. Zou, M. Jung; Th. Schmidt; R. Fink; E. Umbach; J. Chem. Phys., Vol. 121, p.10260, 2004
To clarify the C1s BE for the PDIF-CN2 molecule, we modified the text in lines 216-220 according to the Reviewer suggestion and introduced these new references: “C1s core level (Figure 3a, top curve) shows the contribution from the aromatic backbone shifted at 285.8 eV, a higher BE to PDI8-CN2 as observed in case of F substitution in lateral chains [45,46] and of other electron attracting groups [47,48], and a larger FWHM (1.81 instead of 1.27 eV) due to merging of the C=C and C-H contributions.”
To further strengthen our claims, the BE of C1s in cyano group agrees with reported and expected values, i.e., about 286.8 eV for both PDI8-CN2 and PDIF-CN2 molecules, as reported for other analysis on PDI8-CN2 in Ref Ref 42 and the following new reference added as Ref. 43:
- Self-assembling of cyano- and carboxyl-terminated monolayers using short-chain alkylsiloxane, Zhe Kong, Qi Wang, Liang Ding, Applied Surface Science 256 (2009) 1372–1376
Due to the different chemical environment of backbone carbons in PDIF-CN2, the nitrile CN peak is at about 286.8, i.e., 1.8 eV from a typical C1s 285.0 eV BE but not from our broad main C1s peak.
Concerning the N1s photoemission peak in the nitrile group, we agree with the Referee about its BE; it should be around 399.5-399.7, and, indeed, we found 399.6 eV for both PDI8-CN2 (lines 210-211) and PDIF-CN2 (lines 221-223) thick films.
Why your graphene is so oxydized?
We agree with the Referee about the graphene chemical properties, CO peak is about 14% of the total C1s area (see Figure S2a). This is probably related to the presence of defects due to the transfer process of CVD graphene onto SiO2/Si substrate, as we detected in AFM analysis (see Figure 1c). However, it can be considered very different from graphene oxide, as shown in Ref 49. We agree with the Referee; a clarifying sentence is necessary. We added the following text: “Although we found the presence of a C1s C-O peak, this is probably related to the graphene defects created in the transfer process from CVD synthesis, and our substrate has completely different characteristics from graphene oxide [49].”
Please put C1s and O1s spectra for layers with thickness evolutions.
We agree with the Referee about comparing both molecules' C1s and O1s evolution during growth. Figure 2a and S5b for PDI8-CN2, Figure 3a and S5a for PDIF-CN2 represent the C1s and O1s lineshape evolution vs. thickness, respectively.
N1s position for imides seemed to be too high.
We found a value of about 400.6-400.8 eV for all films and both molecules. It is in good agreement with the value of Ref. 42, references therein, and this new reference, which we added as Ref.44, where a 400.9 eV is reported:
- Solventless polyimide films by vapor deposition; Journal of Vacuum Science & Technology A 4, 369 (1986)
The shift of all peaks F1s and C1s can suggest that the charge compensation is not good.
We thank the Referee for the comment. Our system is not equipped with a charge compensator, so to better clarify this point, we added the following sentence in Section 2, Materials and Methods: “The analysis system is not equipped with a charge compensator, and we never observed energy shifts or peak broadening due to charging during analysis in this study.” As previously claimed, the organic C1s, N1s, and also F1s are in good agreement with reported BEs in similar systems, and the same is for the graphene-related emission. Thus, we did not observe any energy shift due to charging. However, to strengthen our results, we added the following reference (Ref 49) for the CF2 and CF3 BEs:
- XPS Studies of Fluorinated Acrylate Polymers and Block Copolymers with Polystyrene, Camille M. Kassis, Jack K. Steehler, Douglas E. Betts, Zhibin Guan, Timothy J. Romack, Joseph M. DeSimone, and Richard W. Linton; Macromolecules 1996, 29, 3247-3254
2) Many references in the end appear as errors. Fix it please.
We apologize for this issue, and we thank the Referee for this comment; we fixed all crossed references to figures.
3) Are planning to check the stability of your layers?
In this study, our task was to study the growth mechanism, understand the formation of chemical interactions at the organic/graphene interface and finally evaluate the influence of perylenes on the electronic properties of graphene, i.e., tuning its WF and Fermi edge states. We will check the stability of these systems by preparing specific samples, this will require further work, but for some samples, we already checked the electronic properties several months after the deposition. We always found the same results, suggesting good stability in time. To emphasize our future planning, we inserted the following sentence in the Conclusions: “We plan to check our systems' morphologic and electronic properties with time to verify their stability.”
4) Any ideas about scaleup options of your methods? Please add a paragraph for industrual perspectives.
We thank the Referee for this helpful comment. Perylenes are stable and known molecules that can be found easily in commerce. Moreover, the possibility to tune their chemical groups makes these molecules an excellent choice to effectively tune the graphene electronic properties. We used a well-known and established technique to deposit organic molecules, Organic Molecular Beam Deposition, like MBE, fully compatible with most large-scale vacuum production systems. To emphasize this possible development, we added the following sentence in the Conclusions: “The stability, diffusion, and possibility to tune the perylenes’ chemical groups make these organic molecules promising for controlling graphene electronic properties, given possible industrial scale-up, exploiting the OMBD deposition technique compatibility with in-vacuum large-scale production systems. To this aim, we plan to check our systems' morphologic and electronic properties with time to verify their stability.”

Round 2
Reviewer 1 Report
Accept in present form.
Reviewer 2 Report
The authors provided all the required information and answered my questions. However, in the future, I suggest utilizing the XPS instrument equipped with charge compensation when you wish to analyze fine structures of organic molecules.